# Topological phases and bulk-edge correspondence of magnetized cold plasmas

Yichen Fu [1,2,3 ✉] & Hong Qin[1,2,3]

Plasmas have been recently studied as topological materials. However, a comprehensive picture of topological phases and topological phase transitions in cold magnetized plasmas is still missing. Here we systematically map out all the topological phases and establish the bulk-edge correspondence in cold magnetized plasmas. We find that for the linear eigenmodes, there are 10 topological phases in the parameter space of density $n$, magnetic field $B$, and parallel wavenumber $k_z$, separated by the surfaces of Langmuir wave-L wave resonance, Langmuir wave-cyclotron wave resonance, and zero magnetic field. For fixed $B$ and $k_z$, only the phase transition at the Langmuir wave-cyclotron wave resonance corresponds to edge modes. A sufficient and necessary condition for the existence of this type of edge modes is given and verified by numerical solutions. We demonstrate that edge modes exist not only on a plasma-vacuum interface but also on more general plasma-plasma interfaces. This finding broadens the possible applications of these exotic excitations in space and laboratory plasmas.

[1] Princeton Plasma Physics Laboratory, Princeton, NJ, USA. [2] Department of Astrophysical Sciences, Princeton University, Princeton, NJ, USA. [3] These authors contributed equally: Yichen Fu, Hong Qin. ✉email: yichenf@princeton.edu

Recently, the relation between the topological properties of the bulk modes and the chiral (unidirectional) edge modes has attracted growing interest in classical fluid[1–5] and plasma physics[6–10]. Originating in condensed matter physics, the bulk-edge correspondence[11–13] predicts that at the interfaces between two topologically different materials, there exist gapless edge modes across the common band gap. For condensed matter systems, the gap Chern number is a topological invariant for bulk modes to give a $\mathbb{Z}$ classification[12]. On the other hand, continuum media, including plasmas, do not have well defined Brillouin zones, and compactification techniques in the wavenumber space[4,5,10,14] have been adopted to generate integer Chern numbers. It has also been argued that proper Brillouin zones are not essential for using Chern numbers to predicting boundary states[15]. Landau levels in the continuum and Weyl semimetals are good examples.

In plasma physics, the well-known Clemmow-Mullaly-Allis (CMA) diagram[16–19] provides a crude topological classification of wave normal surfaces in the parameter space of magnetic field and density. The classification of the waves using the topological index of the Chern type provides a different theoretical understanding that can be used as predictive tool, e.g., in the study of edge modes. For cold magnetized plasmas, topological phases and topological phase transitions have not been systematically mapped out because of the complicated parameter dependency. The bulk-edge correspondence has not been well established due to the non-compactness of the wavenumber space. For cases with wave vectors perpendicular to the magnetic field, the bulk topology and corresponding edge states of the X waves (transverse-magnetic waves) have been extensively studied[13,14,20–24] when the O waves (transverse-electric waves) are ignored. It is reported that under certain conditions, the bulk-edge correspondence between the gap of X waves can be physically violated[25]. Another type of edge mode has been derived using a simplified analytical model[6] and linked to the Weyl degeneracies[7]. This edge mode was also numerically demonstrated[10] for a plasma-vacuum interface with continuous density falloff. However, the corresponding bulk topological phases and phase transition have not yet been identified.

In the present study, we attempt at a comprehensive picture of the topological phases, topological phase transitions, and the bulk-edge correspondence of magnetized cold plasmas in the absence of a solid boundary. We extend the study by Parker et al.[10] of the linear eigenmodes in a cold plasma with stationary ions in a uniform magnetic field, and carefully draw the topological phase diagram in the parameter space, and clarify the condition for the existence of the topological edge modes. Specifically, we report the following findings. (i) In the parameter space of magnetic field $\mathbf{B} = B_0\hat{z}$, density $n$, and wavenumber $k_z$, there are 10 topological phases, separated by the Langmuir wave-L wave (LL) resonance, the Langmuir wave-Cyclotron wave (LC) resonance, and the $B = 0$ surface. (ii) Their topological properties are classified by the integer Chern numbers of the spectrum. For fixed non-vanishing $B$ and $k_z$, there are two possible topological phase transitions due to the two resonances, while only the transition at the LC resonance produces edge modes. (iii) There exists a critical density $n_c$ such that plasmas below and above $n_c$ are in different topological phases across the LC resonance. We find that edge modes exist not only at the plasma-vacuum boundary[10], but also at more general plasma-plasma interfaces when a necessary and sufficient condition, Eq. (7), is satisfied, and the edge modes can be categorized by different behaviors of the Fermi arcs or Fermi-arcs-like curves. This finding broadens the possible applications of these exotic edge modes in space and laboratory plasmas.

## Results

**Bulk dispersion relation and eigenmodes.** Following ref. [10], we use the linearized fluid equations for a magnetized cold plasma with stationary ion and uniform density $n_e$ to study the eigenmodes of the system. The constant background magnetic field is in the z-direction, i.e., $\mathbf{B}_0 = B_0\hat{z}$, and there is no equilibrium flow. After spacetime Fourier transform, $\partial_t \to -i\omega$, $\nabla \to i\mathbf{k}$, and the governing equations can be written as $H(\mathbf{k})\lvert\psi\rangle = \omega\lvert\psi\rangle$, where $H(\mathbf{k})$ is a $9 \times 9$ Hermitian matrix, and $\lvert\psi\rangle = (\mathbf{v}, \mathbf{E}, \mathbf{B})^\mathrm{T}$ is a nine-dimensional vector consisting of the perturbed velocity and electromagnetic fields. A detailed derivation of $H(\mathbf{k})$ is provided in the "Methods" section.

The dispersion relation is given by the vanishing determinant of $H(\mathbf{k}) - \omega\mathbf{I}_9$, which can be simplified as

$$\det\left(\mathbf{N}\mathbf{N} - N^2\mathbf{I}_3 + \boldsymbol{\epsilon}\right) = 0, \qquad (1)$$

where $\mathbf{N} = c\mathbf{k}/\omega$, and $\boldsymbol{\epsilon}$ is the $3 \times 3$ cold plasma dielectric tensor. In terms of the standard notations in ref. [16],

$$\boldsymbol{\epsilon} = \begin{bmatrix} S & -iD & 0 \\ iD & S & 0 \\ 0 & 0 & P \end{bmatrix}, \qquad P = 1 - \frac{\omega_p^2}{\omega^2}, \qquad (2)$$

$$S = 1 - \frac{\omega_p^2}{\omega^2 - \Omega^2}, \qquad D = -\frac{\Omega}{\omega}\frac{\omega_p^2}{\omega^2 - \Omega^2}, \qquad (3)$$

where $\omega_p = \sqrt{n_e e^2/m_e \epsilon_0}$ is the plasma frequency, and $\Omega = -eB_0/m_e$ is the electron cyclotron frequency. The plasma is called underdense if $\omega_p < |\Omega|$, and overdense if $\omega_p > |\Omega|$. It can be shown that the dispersion relation surfaces in the $(\omega, \mathbf{k})$ space are symmetric with respect to all four coordinate hyperplanes. For each given $k_z$, the system has 9 eigenvalues $\omega_n$ and eigenvectors $\lvert\psi_n\rangle$ as functions of $\mathbf{k}_\perp$, where $\omega_{-n} = -\omega_n$ and $n = -4, -3, ..., 3, 4$ is the index for the eigenmodes. See the "Methods" section for a detailed discussion of the symmetry properties of the spectrum. Note that $\omega_0 = 0$ is the zero frequency eigenmode of the system. The dispersion surfaces $\omega(k_z, k_\perp)$ of the four positive-frequency branches for both overdense and underdense plasmas are shown in Fig. 1a, b. To better illustrate the crossing between branches, the dispersion curves $\omega(k_z)$ at different values of $k_\perp$ are shown in Fig. 1c, d. We can observe that branch crossing occurs only when $k_\perp = 0$, represented by the coldest blue lines. In this case, the horizontal lines $\omega = \omega_p$ are the Langmuir waves given by $P = 0$. The other three branches are the R wave and the L wave given by $N^2 = R$ and $N^2 = L$, respectively, where $R = S + D$ and $L = S - D$. For convenience, we define functions

$$k^\pm := \frac{\omega_p/c}{\sqrt{1 \pm \omega_p/\Omega}}. \qquad (4)$$

When $k_z > 0$, the Langmuir wave resonates with the L wave at $k_z = k^+$. In an underdense plasma, the Langmuir wave also resonates with the lower branch of the R wave, a.k.a electron cyclotron wave, at $k_z = k^-$. Notice that $k^- \to \infty$ when $\omega_p \to |\Omega|$. These four resonant points at $k_z = \pm k^\pm$, previously recognized as the Weyl points[7], play important roles in determining the topological properties of magnetized plasmas.

**Topolgoical phase diagram.** When fixing $k_z$ as a parameter, we can calculate the Chern number for each band, i.e., branch of the dispersion relation, in the $\mathbf{k}_\perp = (k_x, k_y)$ space. If the $\mathbf{k}_\perp$ space can be properly compactified[10,14], Chern numbers in the system should be integers, which are invariant under continuous transforms. It means that each band's Chern number is a topological

invariant that can change only when different bands cross. As shown in Fig. 1, the band crossing in a cold plasma is only possible at the points $(k_x, k_y, k_z) = (0, 0, \pm k^\pm)$ when $B \neq 0$ and $k_z \neq 0$. Therefore, the locations of $k_z = \pm k^\pm$ defines the boundaries between topologically different regions if separated regions have different Chern numbers. In Fig. 2a, the surfaces of $k_z = k^\pm$ and $\Omega = 0$ in the $(\omega_p, \Omega, k_z)$ space are shown, which separate the parameter space into 10 different regions, each represents a different phase characterized by a set of Chern numbers. Although the first band touches the zero-frequency mode at $k_z = 0$, we will show later that this band crossing does not affect the topology. The cross sections of the 3D surfaces at $(\omega_p = 1, \Omega > 0, k_z > 0)$ and

$(\omega_p > 0, \Omega = 1, k_z > 0)$ are shown in Fig. 2b, c, each of which is separated into three phases. Notice that phase I in these cross sections only exists in underdense plasmas, i.e, when $\omega_p < |\Omega|$.

We adopt the same formalism of Berry curvature and regularization strategy used in ref. [10] to calculate the Chern numbers numerically. The Chern number of the $n$-th band is denoted by $C_n$. As functions of $(\omega_p, \Omega, k_z)$, Chern numbers admit the following symmetries, $C_n(\Omega, k_z) = C_n(\Omega, -k_z) = -C_n(-\Omega, k_z) = -C_{-n}(\Omega, k_z)$. See additional details about the symmetry properties of the Chern numbers for the system in the Methods section. Therefore, it suffices to calculate the Chern numbers for the three phases in the domain of $k_z > 0$ and $\Omega > 0$, as shown in Fig. 2b, c.

The resulting Chern numbers for each positive-frequency band in all three phases are shown in Fig. 3a–c. We find that the Chern numbers in phases II and III are $(C_1, C_2, C_3, C_4) = (-1, 1, 1, -1)$ and $(-1, 2, 0, -1)$, respectively, as reported in ref. [10]. For the special case of $k_z = 0$, the Chern numbers for phase III are consistent with those calculated in refs. [14,22]. For phase I, we discover that its Chern numbers are $(C_1, C_2, C_3, C_4) = (0, 0, 1, -1)$, which was not reported previously. When the parameters cross the boundary of $k_z = k^-$ and change from phase I to phase II, bands 1 and 2 cross at $k_\perp = 0$ and change their Chern numbers from $(0, 0)$ to $(-1, 1)$. This change agrees with the fact that the Weyl point at $k_z = k^-$ has Chern number 1[7]. Similar behavior is observed between bands 2 and 3 when parameters cross the boundary of $k_z = k^-$ and change from phase II to phase III.

Of particular interest in the present study is the transition between phases I and II. The boundary between them, i.e., the LC resonance surface, defines a critical density $n_c$, which, expressed in terms of the corresponding plasma frequency $\omega_{p,c} = \sqrt{n_c e^2 / m_e \epsilon_0}$, is

$$\omega_{p,c} = \frac{|\Omega|}{2} \left[ \sqrt{\left(\frac{ck_z}{\Omega}\right)^4 + 4\left(\frac{ck_z}{\Omega}\right)^2} - \left(\frac{ck_z}{\Omega}\right)^2 \right]. \quad (5)$$

For fixed $k_z$ and $\Omega$, transition between phases I and II occurs at $n = n_c$. Notice that when $k_z \to \infty$, $\omega_{p,c} \to |\Omega|$, and $n_c$ becomes $n_c^\infty \equiv \Omega^2 m_e \epsilon_0 / e^2$.

The surface of $\Omega = 0$ is also a boundary between different topological phases because $C_n(-\Omega) = -C_n(\Omega)$. For the special

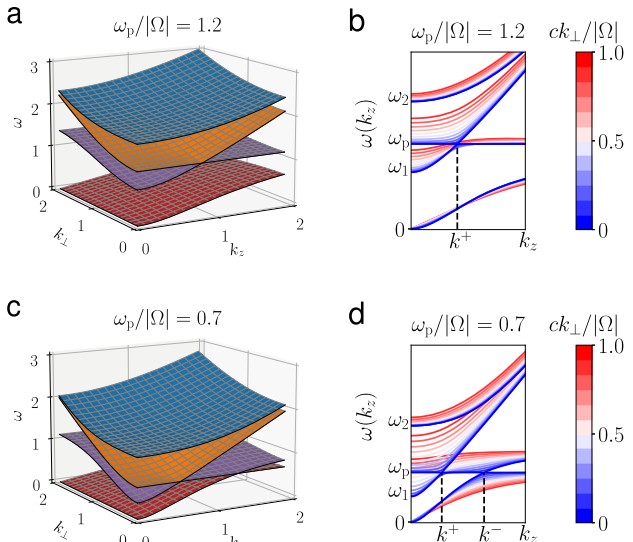

**Fig. 1 Dispersion relations of magnetized cold plasmas. a** Dispersion relation surfaces of an overdense plasma. Only the $k_\perp > 0$ and $k_z > 0$ part of positive-frequency branches are shown. Different colors represent different branches. **b** The dispersion curves $\omega(k_z)$ at fixed $k_\perp$ of an overdense plasma. The colors represent different values of $k_\perp$. Only the crossing of the curves with the same color indicates the crossing of different branches. $\omega_{1,2}/|\Omega| = (\sqrt{4r^2 + 1} \pm 1)/2$, where $r = \omega_p/|\Omega|$. $k^\pm$ are given by Eq. (4) assuming $\Omega > 0$. (**c**), (**d**) Same as (**a**), (**b**) but for an underdense plasma.

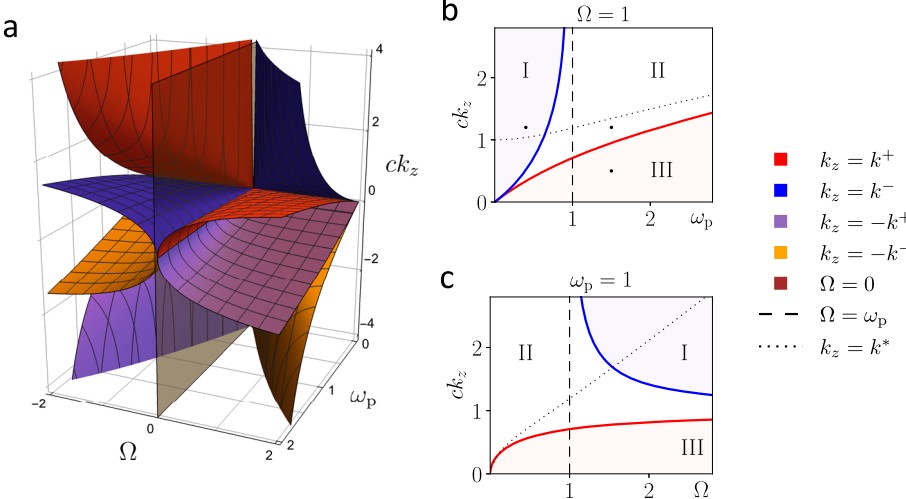

**Fig. 2 Topological phase diagrams of magnetized cold plasma. a** 3D phase diagram of a cold plasma in the $(\omega_p, \Omega, k_z)$ space. There are 10 topological phases. **b**, **c** 2D cross sections of (**a**) at $\omega_p = 1$ and $\Omega = 1$. Only the $k_z > 0$ and $\Omega > 0$ part is shown. Dashed lines indicates $\Omega = \omega_p$. Dotted lines indicates $k_z = k^*$. The Roman numerals I-III indicate three different topological phases in each cross section. The band structures and Chern numbers at three black dots in (**b**) are shown in Fig. 3.

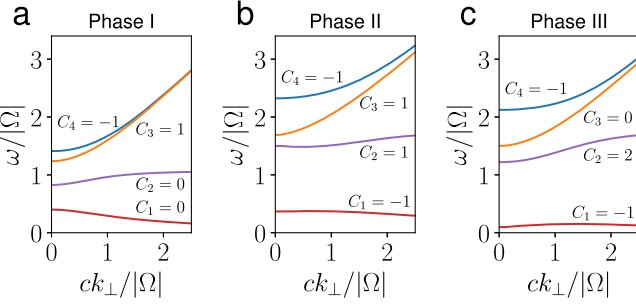

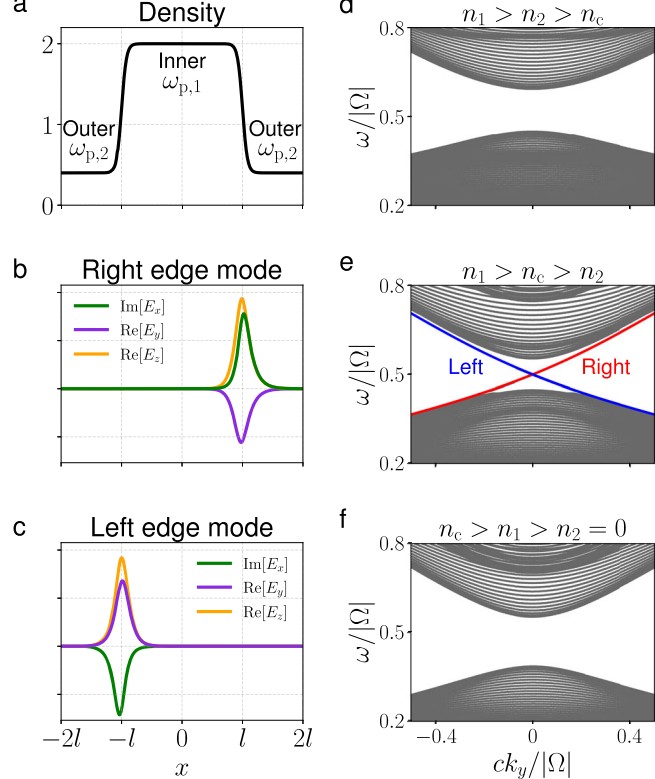

**Fig. 3 Chern numbers. a–c** Chern numbers of each positive frequency bands of the three phases of the phase diagram. Since the dispersion relation is isotropic in $(k_x, k_y)$ plane, only the $k_\perp > 0$ part is shown. The parameters used are $\Omega = 1$ and $(\omega_p, ck_z) = (0.4, 1.2)$ in (**a**), $(1.5, 1.2)$ in (**b**), and $(1.5, 0.5)$ in (**c**), which are shown as black dots in Fig. 2b.

case of $k_z = 0$, the boundary between phases I and II and the boundary between phases II and III collapse to the lines defined by $\Omega = 0$ and $\omega_p = 0$. In this case, the only possible nontrivial phase transition happens at the $\Omega = 0$ surface.

**Band gaps**. Bulk-edge correspondence suggests that edge modes exist in the common band gaps at the interface between two topological materials with different gap Chern numbers. We now identify possible band gaps in a magnetized cold plasma. At fixed $k_z$, when $k_\perp \to \infty$, $(\omega_1, \omega_2, \omega_3, \omega_4) \to (0, \omega_{uh}, ck_\perp, ck_\perp)$, where $\omega_{uh}^2 = \omega_p^2 + \Omega^2$ is the upper hybrid frequency. Thus, when $|k_z| \neq k^\pm$, it is possible to have gaps between bands 1 and 2 and between bands 2 and 3. However, the gap between bands 2 and 3 does not always exist. When $k_\perp = 0$, $\omega_3(k_z)$ is given by $N^2 = L$. The non-overlapping of bands 2 and 3 requires $\omega_3(k_z) > \omega_{uh}$, which leads to

$$|k_z| > k^* := \frac{|\Omega|}{c}\left(1 + \frac{\omega_p^2}{\Omega^2}\right)^{1/4}. \qquad (6)$$

The locations of $|k_z| = k^*$ in the parameter space are shown in Fig. 2b, c. It is clear that the gap between bands 2 and 3 does not exist in phase III because condition (6) is not satisfied there.

The topology of a band gap is characterized by it gap Chern number, which is defined as $C_{i,i+1} = \sum_{n=-4}^{i} C_n$ for the gap between the $i$-th and $(i+1)$-th bands. In phase I, both $C_{1,2}$ and $C_{2,3}$ are trivially zero, as in the phase of vacuum[11] that phase I neighbors at the boundary of $\omega_p = 0$. Thus, as far as the gap topology is concerned, phase I plasmas are identical to the vacuum, which is interesting if not surprising. In phase II, the gap Chern numbers $(C_{1,2}, C_{2,3})$ become $(-1, 0)$, indicating a topological phase transition at the boundary between phases I and II due to the crossing of the gap between bands 1 and 2. In phase III, the gap Chern numbers $(C_{1,2}, C_{2,3})$ are $(-1, 1)$. Although $C_{2,3}$ is different between phase II and III, there is no band gap between bands 2 and 3 in phase III as proved above. When $|k_z| \neq 0$, there is another band gap between bands 0 and 1. However, the gap Chern number $C_{0,1}$ for this gap is zero for all three phases, and it is a trivial band gap. Therefore, only the band gap between bands 1 and 2 shared by phases I and II is interesting in the context of bulk-edge correspondence.

It is worth mentioning that when $k_z = 0$, band 3 becomes the O wave, and bands 2 and 4 are the X wave. If one chooses to ignore the O wave[13,14,20,22], then a band gap shows up between bands 2 and 4, as long as $\Omega \neq 0$. The physical properties of this gap has been extensively studied, including the violation of bulk-edge correspondence under certain conditions[25]. However, as an important eigenmode in magnetized cold plasmas, band 3 always exists. Especially when $|k_z| > k^+$, band 3 has a non-zero Chern

**Fig. 4 The band structures of nonuniform plasmas at $ck_z/|\Omega| = 0.7$.** **a** Schematic diagram of the density profile in $x$ direction, where $cl/|\Omega| = 40$ and $c\delta/|\Omega| = 4$. **b, c** The non-zero components of electric fields of edge modes in (**e**) at $ck_y/|\Omega| = 0.05$. **d–f** The band structure $\omega(k_y)$ with various inner and outer densities. The plasma frequencies $(\omega_{p,1}, \omega_{p,2})/|\Omega|$ are $(0.8, 0.6)$, $(0.6, 0.4)$, and $(0.4, 0)$ in (**d**)–(**f**), respectively. The topological edge modes on left and right side are shown by blue and red lines, respectively. The rest modes are shown by gray lines. Notice that the critical plasma frequency given by Eq. (5) is $\omega_{p,c}/|\Omega| \approx 0.5$.

number and should not be ignored. In the present study, we include all 9 bands in magnetized cold plasmas.

**Bulk-edge correspondence**. Having established the topological phase diagram and possible band gaps in a magnetized cold plasma, we now investigate the edge modes at the interface between two different magnetized cold plasmas. As discussed above, at a fixed $B$, the only possible nontrivial gap that admits two different gap Chern numbers is the gap between bands 1 and 2 shared by phases I and II. The bulk-edge correspondence then predicts that edge modes exist in the common band gap at the interface between a phase I plasma and a phase II plasma. We now solve for these edge modes numerically in an 1D inhomogeneous plasma. The background magnetic field is constant, i.e., $\mathbf{B}_0 = B_0\hat{z}$, and the plasma density is nonuniform only in the $x$-direction, as shown in Fig. 4a. The density profile is given by $n(x) = \frac{1}{2}(n_1 - n_2)\{\tanh[-(x - l)/\delta] + \tanh[(x + l)/\delta]\} + n_2$, where $n_1$ and $n_2$ are the densities of the inner and outer plasmas, and $l$ and $\delta$ are the location and width of the interface. For realistic plasmas, the width of the interface is finite, i.e., $\delta > 0$.

Because the system is uniform in the $z$-direction, the parallel wavenumber $k_z$ enters as a parameter. The inner and outer plasmas can be represented by two points, $(\omega_{p,1}, \Omega, k_z)$ and $(\omega_{p,2}, \Omega, k_z)$, in the phase diagram shown in Fig. 2. Here, $\omega_{p,1}$ and $\omega_{p,2}$ are the inner and outer plasma frequencies. As discussed above, when there is a common band gap shared by the inner and outer plasmas, chiral

edge modes exist in the gap if and only if the inner plasma is in phase I and outer plasma is in phase II such that they have different gap Chern numbers. Furthermore, the number of chiral edge modes at the interface should be equal to the difference of the gap Chern numbers, which is 1 in the present case. Since $n_c$ is the critical density at the boundary between phases I and II, for given $k_z$ and $\Omega$, the chiral edge mode exists if and only if

$$n_1 > n_c > n_2, \tag{7}$$

which, expressed in terms of plasma frequencies, is

$$\frac{\omega_{p,1}}{|\Omega|} + \frac{\omega_{p,1}^2}{c^2 k_z^2} > 1 > \frac{\omega_{p,2}}{|\Omega|} + \frac{\omega_{p,2}^2}{c^2 k_z^2}. \tag{8}$$

Here, we observe that underdense and overdense plasmas behave differently. When both the inner and outer plasmas are overdense, edge mode cannot exist regardless of $k_z$. If the inner plasma is overdense while the outer plasma is underdense, edge modes can be found when $k_z > k^-(\omega_{p,2})$. If both the inner and outer plasma are underdense, the edge mode can only be found when $k^-(\omega_{p,1}) > |k_z| > k^-(\omega_{p,2})$. Notice that when the outer side is a vacuum, the criteria given by Eq. (8) can be satisfied in two scenarios, either $k_z^2$ is small enough or the inner plasma is overdense. Incidentally, all the parameters chosen in ref. [10] belong to the first scenario.

To numerically verify the criteria in Eqs. (7) or (8), we Fourier-transform in $y$, $z$ and $t$, then spatially discretize the Hamiltonian $H$ in the $x$-direction using a finite difference method. A periodic boundary condition at $x = \pm 2l$ is adopted, as in ref. [1]. To ensure the numerically calculated spectrum admits the same symmetries as the analytical spectrum, we adopt a discretization scheme that preserves the Hermiticity and the particle-hole symmetry of the system. See the "Methods" section for details. The structures of the positive-frequency bands for different $(n_1, n_2)$ at $ck_z/|\Omega| = 0.7$ are shown in Fig. 4d–f. For the cases of $n_c > n_1 > n_2 = 0$ and $n_1 > n_2 > n_c$, there is no edge mode in the gap between the first and second bands. In particular, Fig. 4f shows that the edge modes can be absent at a plasma-vacuum interface. In Fig. 4e, $n_1 > n_c > n_2$ and there are two gapless edge modes in the band gap. Due to the symmetry of $\omega(k_y) = \omega(-k_y)$, the two edge modes cross at $k_y = 0$. The electric field structure of the two gapless modes localized at different edges is shown in Fig. 4b, c. The number of edge modes at each edge is the same as the difference of the gap Chern number, consistent with the prediction of bulk-edge correspondence.

To further understand the edge modes, band structures at $k_y = 0$ with various densities are plotted in Fig. 5, which shows that edge modes can be classified by behaviors of the Fermi arc or Fermi-arc-like curves. The inner plasma is underdense in Fig. 5a, c and overdense in Fig. 5b, d. The outer is underdense in Fig. 5c, d and vacuum in Fig. 5a, b. The bulk dispersions are shown by orange and blue lines for inner and outer plasmas in each case. The topological edge modes are shown in red lines representing the crossing points between the left and right edge modes at each given $k_z$. We can see that the range where edge modes exist coincides with Eq. (8) exactly. Noticeably, in Fig. 5a, edge modes exist when $|k_z| < k^-(\omega_{p,1})$ between a underdense plasma and a vacuum. The dispersion surface $\omega = \omega(k_z, k_y)$ of the edge modes connects the two Weyl points of the inner plasma at $k_z = \pm k^-(\omega_{p,1})$. The intersection of this dispersion surface and $\omega = \omega_{p,1}$ is known as the Fermi arc connecting two Weyl points[26,27]. When the inner plasma is overdense in Fig. 5b, the Weyl points $\pm k^-(\omega_{p,1})$ disappear, but the edge modes still exist and the dispersion surface connects to $k_z = \pm\infty$. When the inner and outer plasmas are all underdense in Fig. 5c, edge modes are prohibited if $-k^-(\omega_{p,2}) < k_z < k^-(\omega_{p,2})$, then the dispersion surface no longer connects the Weyl points of the inner plasma at positive and

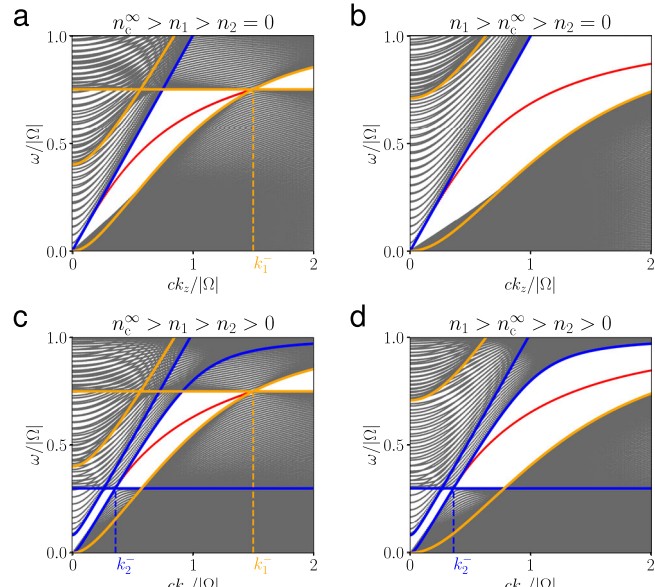

**Fig. 5 Four possible types of Fermi-arc-like structures. a–d** Band structure $\omega(k_z)$ at $k_y = 0$ with different parameters, where Fermi-arc-like structures of edge mode connecting Weyl points. The topological edge modes are highlighted by the red curves. The bulk modes of the inner and outer plasmas when $k_\perp = 0$ are highlighted by orange and blue curves, respectively. The other bulk modes are indicated by gray curves. $k_1^- = k^-(\omega_{p,1})$ and $k_2^- = k^-(\omega_{p,2})$. The plasma frequencies $(\omega_{p,1}, \omega_{p,2})/|\Omega|$ in (**a**)–(**d**) are (0.75, 0), (1.1, 0), (0.75, 0.3), and (1.1, 0.3).

negative $k_z$. Instead, a Fermi-arc-like curve connects $k_z = k^-(\omega_{p,1})$ and $k_z = k^-(\omega_{p,2})$, the Weyl points of the inner and outer plasmas. In Fig. 5d, the inner is overdense and the outer underdense, a Fermi-arc-like curve connects the Weyl point of the outer plasma to infinity. These numerical results also confirm the condition given by Eqs. (7) or (8) for the existence of the edge modes.

As a side note, the gap between bands 1 and 2 of the inner plasma may overlap with the gap between bands 2 and 3 of the outer plasma. It is reasonable to suggest that if $C_{in,1,2} \neq C_{out,2,3}$, edge modes might exist within this gap. However, this common gap is filled with other eigenmodes in reality. The upper hybrid frequency $\omega_{uh}$, which depends on plasma density, sets the upper range for band 2. Since the density profile is continuous, the local upper hybrid frequency $\omega_{uh}(x)$ will always fill in the gap between the second bands of the inner and outer plasmas. An example is illustrated in Fig. 6, where the inner and outer plasmas belong to phases II and I, respectively.

Varying density is a convenient but not the only way to create interfaces between topologically different plasmas. Since plasma topology varies with the strength and direction of the magnetic field, one can create topologically nontrivial interfaces by assembling two plasmas with different background magnetic field. In particular, when $k_z = 0$, the only possible topological phase transition in a cold plasma occurs at the $\Omega = 0$ surface, as discussed above. In this case, the edge modes at the interface between two plasmas with opposite magnetic fields has been studied for the X wave[13,23]. However, as a whole system, such a setup is inhomogeneous, and the O wave cannot be decoupled. More thorough analysis is need.

## Discussion

The linear and nonlinear properties of edge modes, or surface waves, in plasmas have been studied for decades[28–30]. However, the relation between surface modes and the topology of bulk

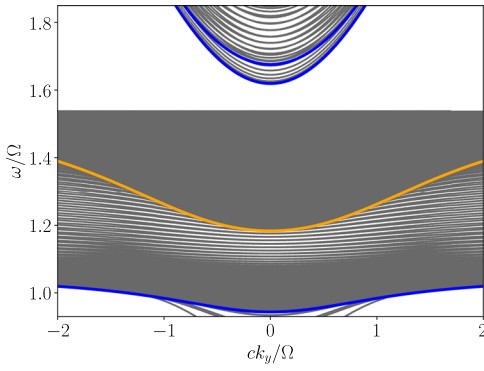

**Fig. 6 Band structure when two different gaps overlap.** In this case, $ck_z/|\Omega| = 1.4$ and the inner and outer plasma frequency are $(\omega_{p,1}, \omega_{p,2})/|\Omega| = (1.4, 0.1)$. The bulk modes of the inner and outer plasma when $k_\perp = 0$ are highlighted by orange and blue curves, respectively. The other bulk modes are indicated by gray curves. The gap between bands 1 and 2 of the inner plasma overlaps with gap between bands 2 and 3 of the outer plasma. However, this band gap is filled with local upper hybrid modes due to continuous density profile.

plasma was only pointed out recently. In the present research, we studied the linear eigenmodes of a cold plasma in a uniform magnetic field, and found that the system has 10 different topological phases in the $(\omega_p, \Omega, k_z)$ space. The different phases are separated by the surfaces of Langmuir wave-L wave resonance, Langmuir wave-cyclotron wave resonance, and $\Omega = 0$. We found that at fixed $\Omega$ and $k_z$, only the band gap between bands 1 and 2 shared by phases I and II is interesting in the context of bulk-edge correspondence, and that the necessary and sufficient condition for the existence of edge modes is $n_1 > n_c > n_2$. These findings were verified by numerical studies of the corresponding chiral edge modes in 1D inhomogeneous plasmas. The edge modes exist not only on the plasma-vacuum interface, but also on more general plasma-plasma interfaces. The validity of the bulk-edge correspondence is confirmed for the cold magnetized plasma as a non-driven Hermitian system. These results improved our understanding of the elementary properties of the magnetized plasma as a topological material.

It is worth mentioning that in the plasma physics community, the term "edge" usually refers to the physical boundary of plasmas adjacent to the first wall of vacuum chambers[31,32]. In the context of topological matters, the term 'edge' refers to the boundary of two topologically different regions, although a plasma-vacuum edge can also be a topological edge when the plasma and the vacuum have different Chern numbers. As discussed above, the topological edges include not only the plasma-vacuum boundary, but also more general gaseous plasma-plasma interfaces, which are the focus of the present study We did not consider gaseous plasma-solid interfaces, which, unlike the interfaces between solid state materials[27], involve more complex physical processes, such as the plasma sheath and plasma-wall interactions[31,33]. It is not appropriate to model gaseous plasma-solid interfaces only as simple interfaces between two different topological materials.

The present classification of the topological phases in the parameter space is carried out for the linearized system of a homogeneous, magnetized, cold plasma with stationary ions. Such a classification in this simplified system provides an elementary tool and serves as a reference model for studying the topological-matter properties of more realistic plasmas. For example, in an inhomogeneous plasma, edge modes may be excited at the interface between two plasmas in different topological phases. When other physical effects, such finite temperature, plasma collisions, and kinetic interactions, are important,

the band structures[16] and topological properties of the system can change significantly, including the number of topological phases and possible phase transitions. One systematic approach for understanding these changes is to analyze the variations of the symmetry properties of the system[34]. For instance, the magnetized cold plasma model studied here has a broken time-reversal symmetry, but the linearized ideal magnetohydrodynamics system is invariant under the (modified) time-reversal transformation[8], despite the existence of an external magnetic field. Furthermore, the linear dynamics in many plasma models is expected be to non-Hermitian[35,36], permitting unstable and damped eigenmodes. Applying the methods of topological phases for non-Hermitian systems[37,38] will bring more insights and discoveries in the study of plasma instabilities for laboratory and astrophysical plasmas.

## Methods

**Basic equations**. Here we outline the band structure of cold plasma waves. Assume the background plasma is uniform and stationary, the ions are motionless and the magnetic field is constant, i.e., $\mathbf{B}_0 = B_0\hat{z}$. The linearized fluid equations are[16]

$$\partial_t \mathbf{v} = \frac{e}{m_e}(\mathbf{E} + \mathbf{v} \times \mathbf{B}_0),$$
$$\partial_t \mathbf{E} = c^2 \nabla \times \mathbf{B} - \frac{e n_e}{\epsilon_0}\mathbf{v}, \qquad (9)$$
$$\partial_t \mathbf{B} = -\nabla \times \mathbf{E},$$

where $\mathbf{v}, \mathbf{B}, \mathbf{E}$ are perturbed velocity, magnetic field and electric field, $e$ is the electron charge, $m_e$ and $n_e$ are electron mass and density, and $c$ is light speed. Let plasma frequency $\omega_p^2 = n_e e^2/\epsilon_0 m_e$ and cyclotron frequency $\Omega = -eB_0/m_e$. Define renormalized velocity $\tilde{\mathbf{v}} = \omega_p \mathbf{v}$, reference electric field $\bar{E}$ and reference frequency $\bar{\omega}$. We normalize time to $\bar{\omega}^{-1}$, frequency to $\bar{\omega}$, length to $c/\bar{\omega}$, $\tilde{\mathbf{v}}$ to $e\bar{E}/m_e$, electric field to $\bar{E}$, and magnetic field to $\bar{E}/c$. The equation system becomes

$$\partial_t \tilde{\mathbf{v}} = \omega_p \mathbf{E} - \Omega \tilde{\mathbf{v}} \times \hat{z},$$
$$\partial_t \mathbf{E} = \nabla \times \mathbf{B} - \omega_p \tilde{\mathbf{v}}, \qquad (10)$$
$$\partial_t \mathbf{B} = -\nabla \times \mathbf{E}.$$

Notice that when the density is not uniform, it suffices to change $\omega_p$ to $\omega_p(\mathbf{r})$ in Eq. (10). From now on, we omit the tilde for convenience. After space Fourier transform, $\partial_t \rightarrow -i\omega$, $\nabla \rightarrow i\mathbf{k}$, and the governing equations can be written as $H|\psi\rangle = \omega|\psi\rangle$, where $|\psi\rangle = (\mathbf{v}, \mathbf{E}, \mathbf{B})^T$ and

$$H(\omega_p, \Omega, \mathbf{k}) = \begin{pmatrix} i\Omega\hat{z}\times & i\omega_p & 0 \\ -i\omega_p & 0 & -\mathbf{k}\times \\ 0 & \mathbf{k}\times & 0 \end{pmatrix}. \qquad (11)$$

Here, $H$ is a $9 \times 9$ Hermitian matrix. The dispersion relation is given by $\det(H - \omega I_9) = 0$, which simplifies to Eq. (1).

**Symmetries of the system**. As a real system, the equations of motion for plasmas admit an unbreakable particle-hole symmetry[9,39]. For our system, the symmetry states that $H(-\mathbf{k})^* = -H(\mathbf{k})$, where $^*$ denotes complex conjugate. It ensures that the dispersion relation has the symmetry of $\omega(-\mathbf{k}) = -\omega(\mathbf{k})$. Notice that Eq. (1) remains invariant under $\mathbf{k} \rightarrow -\mathbf{k}$, so the dispersion relation satisfy $\omega(\mathbf{k}) = \omega(-\mathbf{k})$ as well. In addition, since the system is isotropic in the direction perpendicular to the background magnetic field, $\omega$ only depends on $k_z \equiv k_z$ and $k_\perp \equiv \sqrt{k_x^2 + k_y^2}$. Therefore, the dispersion relation surfaces in the $(\omega, k_x, k_y, k_z)$ space are symmetric with respect to the reflections of all four coordinate hyperplanes.

The symmetries of eigenvalues can also be obtained, which will be useful during the calculation of the Chern numbers. $H(\omega_p, \Omega, \mathbf{k})$ has 9 eigenvalues, one of which is identically zero. The eigenvalues and corresponding eigenvectors can be labeled as $\omega_n, |\psi_n\rangle$, where $n = -4, -3, \cdots, 3, 4$, $\omega_i < \omega_j$ if $i < j$. Assume that at some $(\omega_p, \Omega, \mathbf{k})$, the $n$-th eigenvalue and eigenvector are $\omega_n = \omega$ and $|\psi_n\rangle = (\mathbf{v}, \mathbf{E}, \mathbf{B})^T$. We can verify the following symmetries of eigenvalues and eigenvectors:

1. For the reflection of band number $n \rightarrow -n$, $\omega_{-n} = -\omega$ and $|\psi_{-n}\rangle = (\mathbf{v}^*, \mathbf{E}^*, -\mathbf{B}^*)^T$.
2. For the reflection of magnetic field $\Omega \rightarrow -\Omega$, $\omega_n = \omega$ and $|\psi_n\rangle = (-\mathbf{v}^*, \mathbf{E}^*, \mathbf{B}^*)^T$.
3. For the reflection of wavenumber $\mathbf{k} \rightarrow -\mathbf{k}$, $\omega_n = \omega$ and $|\psi_n\rangle = (\mathbf{v}, \mathbf{E}, -\mathbf{B})^T$.

**The symmetries of Chern numbers**. In this section, we briefly describe the calculation and symmetries of Chern numbers. For any given parallel wavenumber $k_z$,

the Chern number can be calculated in the $\mathbf{k}_\perp$ space for each band by[12] $C_n = (2\pi)^{-1} \int d\mathbf{S} \cdot \mathbf{F}_n(\mathbf{k})$, where $\mathbf{F}_n = \nabla_\mathbf{k} \times \mathbf{A}_n$ is the Berry curvature, and $\mathbf{A}_n = i\langle\psi_n|\nabla_\mathbf{k}\psi_n\rangle$ is the Berry connection. Let $\left|\psi_n(\omega_\mathrm{p}, \Omega, \mathbf{k})\right\rangle = (\mathbf{v}, \mathbf{E}, \mathbf{B})^\mathrm{T}$. The Berry connection is $\mathbf{A}_n(\omega_\mathrm{p}, \Omega, \mathbf{k}) = i(\mathbf{v}^\dagger\nabla_\mathbf{k}\mathbf{v} + \mathbf{E}^\dagger\nabla_\mathbf{k}\mathbf{E} + \mathbf{B}^\dagger\nabla_\mathbf{k}\mathbf{B})$, where $\dagger$ denotes conjugate transpose. Based on the symmetries of eigenvectors, we obtain the following symmetries of Chern numbers:

1. For the reflection of band number $n \to -n$,

$$\begin{aligned} \mathbf{A}_{-n}(\omega_\mathrm{p}, \Omega, \mathbf{k}) &= i\langle\psi_{-n}|\nabla_\mathbf{k}\psi_{-n}\rangle \\ &= i(\mathbf{v}\nabla_\mathbf{k}\mathbf{v}^\dagger + \mathbf{E}\nabla_\mathbf{k}\mathbf{E}^\dagger + \mathbf{B}\nabla_\mathbf{k}\mathbf{B}^\dagger) \\ &= -\mathbf{A}_n(\omega_\mathrm{p}, \Omega, \mathbf{k}). \end{aligned} \qquad (12)$$

Thus, $C_{-n}(\omega_\mathrm{p}, \Omega, k_z) = -C_n(\omega_\mathrm{p}, \Omega, k_z)$.

2. For the reflection of magnetic field $\Omega \to -\Omega$, $\mathbf{A}_n(\omega_\mathrm{p}, -\Omega, \mathbf{k}) = -\mathbf{A}_n(\omega_\mathrm{p}, \Omega, \mathbf{k})$. Thus, $C_n(\omega_\mathrm{p}, -\Omega, k_z) = -C_n(\omega_\mathrm{p}, \Omega, k_z)$.

3. For the reflection of parallel wavenumber $k_z \to -k_z$, since the system is isotropic in $\mathbf{k}_\perp$-plane, it is equivalent to the reflection of wavenumber $\mathbf{k} \to -\mathbf{k}$. Due to the symmetries of the eigenvectors, we have $\mathbf{A}_n(\omega_\mathrm{p}, \Omega, -\mathbf{k}) = -\mathbf{A}_n(\omega_\mathrm{p}, \Omega, \mathbf{k})$ and $\mathbf{F}_n(\omega_\mathrm{p}, \Omega, -\mathbf{k}) = \mathbf{F}_n(\omega_\mathrm{p}, \Omega, \mathbf{k})$. Therefore, $C_n(\omega_\mathrm{p}, \Omega, -k_z) = C_n(\omega_\mathrm{p}, \Omega, k_z)$.

For numerical evaluation of the Chern numbers, the following alternative formula of Berry curvature is used,

$$\mathbf{F}_n = i\sum_{m\neq n}\frac{\langle\psi_n|\nabla_\mathbf{k}H|\psi_m\rangle \times \langle\psi_m|\nabla_\mathbf{k}H|\psi_n\rangle}{(\omega_m - \omega_n)^2}. \qquad (13)$$

To enforce integer Chern numbers, we adopt the same regularization strategy used in ref. [10]. At large $k_\perp$, we regularize the plasma frequency in Eq. (11) by replacing $\omega_\mathrm{p}$ with $\omega_\mathrm{p}/(1 + k_\perp^2/k_\mathrm{c}^2)$, where $k_\mathrm{c}$ is a large-enough cutoff wavenumber.

**Numerical scheme**. Here we introduce the numerical methods for eigenmodes calculation. When density $n(x)$ is nonuniform in the $x$-direction, we Fourier-transform Eq. (10) in $y, z, t$ and but not in $x$. The simulation region in the $x$-direction is $[-2l, 2l]$ and is discretized into $N$ grids. The interval of grids is $\Delta x = 4l/N$ and the grid points are $x_i = i\Delta x - 2l$, $i = 0, \cdots, N-1$. Similar to ref. [1], a periodic boundary condition is applied at $x = \pm 2l$. Next, we adopt the strategy of structure-preserving geometric algorithms in plasma physics to discretize $\mathbf{v}$ and $\mathbf{E}$ on integer grid points $x_i$ and $\mathbf{B}$ on half-integer grid points $x_{i+1/2} \equiv (x_i + x_{i+1})/2$, i.e., $\mathbf{v}_i \equiv \mathbf{v}(x_i)$, $\mathbf{E}_i \equiv \mathbf{E}(x_i)$, and $\mathbf{B}_{i+1/2} \equiv \mathbf{B}(x_{i+1/2})$. Such discretization ensures centered discretization of $x$-derivatives and preserves the geometric relations between different components of the field. Periodic boundary condition enforces that $\mathbf{v}_N = \mathbf{v}_0$, $\mathbf{E}_N = \mathbf{E}_0$, $\mathbf{B}_{N+1/2} = \mathbf{B}_{1/2}$. Define $\omega_{\mathrm{p},j} \equiv \omega_\mathrm{p}(x_j) \sim \sqrt{n(x_j)}$. Then, Eq. (10) is discretized as

$$\begin{cases} \omega\, v_{x,i} = -i\Omega v_{y,i} + i\omega_{\mathrm{p},i}E_{x,i}, \\ \omega\, v_{y,i} = i\Omega v_{x,i} + i\omega_{\mathrm{p},i}E_{y,i}, \\ \omega\, v_{z,i} = i\omega_{\mathrm{p},i}E_{z,i}. \end{cases} \qquad (14)$$

$$\begin{cases} \omega\, E_{x,i} = -i\omega_{\mathrm{p},i}v_{x,i} + k_z\frac{B_{y,i+1/2}+B_{y,i-1/2}}{2} - k_y\frac{B_{z,i+1/2}+B_{z,i-1/2}}{2}, \\ \omega\, E_{y,i} = -i\omega_{\mathrm{p},i}v_{y,i} - k_z\frac{B_{x,i+1/2}+B_{x,i-1/2}}{2} - i\frac{B_{z,i+1/2}-B_{z,i-1/2}}{\Delta x}, \\ \omega\, E_{z,i} = -i\omega_{\mathrm{p},i}v_{z,i} + k_y\frac{B_{x,i+1/2}+B_{x,i-1/2}}{2} + i\frac{B_{y,i+1/2}-B_{y,i-1/2}}{\Delta x} \end{cases} \qquad (15)$$

$$\begin{cases} \omega\, B_{x,i+1/2} = -k_z\frac{E_{y,i+1}+E_{y,i}}{2} + k_y\frac{E_{z,i+1}+E_{z,i}}{2}, \\ \omega\, B_{y,i+1/2} = k_z\frac{E_{x,i+1}+E_{x,i}}{2} + i\frac{E_{z,i+1}-E_{z,i}}{\Delta x}, \\ \omega\, B_{z,i+1/2} = -k_y\frac{E_{x,i+1}+E_{x,i}}{2} - i\frac{E_{y,i+1}-E_{y,i}}{\Delta x}. \end{cases} \qquad (16)$$

Equations (14)–(16) are now a standard matrix eigenvalue problem, which can be solved by various established algorithms. It is straightforward to confirm that the matrix specified by Eqs. (14)–(16) is Hermitian and admits the particle-hole symmetry as Eq. (10) is and does. This structure-preserving discretization enables the reformulation of the eigenvalue problem of an inhomogeneous, magnetized, cold plasma as an eigenvalue problem for a Hermitian matrices with the particle-hole symmetry.

## Data availability
All relevant data supporting the findings of this study are available on DataSpace at Princeton University (http://arks.princeton.edu/ark:/88435/dsp01qj72pb21d) or from the authors on request.

## Code availability
The computer code used to generate results that are reported in the paper is available from the authors on request.

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

## Acknowledgements

This research was supported by the U.S. Department of Energy (DE-AC02-09CH11466). We thank Jeff Parker and Fang Xie for fruitful discussions.

## Competing interests
The authors declare no competing interests.
