## [Peer Review File · Nature Communications]

Reviewers' Comments:

Reviewer #1:

Remarks to the Author:

Report on :

Title : "Topological phases, topological phase transition, and bulk-edge correspondence of magnetized cold plasmas"

Authors : Yichen Fu and Hong Qin

This work provides a comprehensive analysis of the topological phase point of view applied to cold plasma waves classification in a magnetized plasma. When the density, magnetic field and parallel wave number are varied, the various branches of the cold magnetized plasma dispersion display 10 topological phases. Then, on the basis of this identification and classification, the bulk-edge correspondence is used to present four edge modes expected from this bulk-edge correspondence. The issue of magnetized plasma waves classification has already been addressed in the past but this new, global and clear view on the subject, the radically innovative methods as well as the new results presented in this paper deserve publication in Nature Com. The new findings presented here broaden the perspective and will be of interest to others communities beside the plasma community.

I really feel that this paper will influence thinking in the field, particularly it will renew the sixty years old presentation of the CMA diagram* providing a crude topological classification of the modes transition in the density/magnetic field space, but on the restricted phenomenological basis of the visual transition between the wave normal surfaces (roughly speaking transition between spheroid, lemniscoid and hyperboloid), here we have at hand a new topological index of the Chern type providing a predictive tool with respect to the occurrence of edge modes.

Beside the bulk mode topological classification, the edge modes identified and analyzed here are interesting both from the point of view of theoretical and applied plasma physics and will clarify the Dyakonov** type surface waves classification at the interface between magnetized plasmas.

* CMA : P.C. Clemmow and R.F. Mullaly 1953 and W.P. Allis, Sherwood 1959.

** M.I. Dyakonov Zh. Eksp. Teor. Fiz. 94, 119-123 (April 1988).

Reviewer #2:

Remarks to the Author:

The manuscript "Topological phases, topological phase transition, and bulk-edge correspondence of magnetized cold plasmas" by Fu and Qin sets out to study an interesting problem and I was anticipating an interesting read. However, the content was more standard and less intriguing than I had hoped. I do not share the authors viewpoint that proper Brillouin zones are needed for using Chern numbers to predicting boundary states. Landau levels are also described in the continuum and the quantum Hall effect is in fact the best established physical application of Chern numbers. Similarly the Chern number of the nodes in Weyl semimetals are related to the Fermi arcs on the boundary and there is no problem describing these phenomena in the continuum.

The authors conclude: "No violation of bulk-edge correspondence is observed in this system." This is more of a sanity check as the system is neither driven nor non-Hermitian. In fact, I would have hoped that authors would have started their exploration where they instead finish the discussion of future directions, namely with the study of non-Hermitian topological phases in plasmas. In that context there is still a lot to learn and explore despite exciting progress the past 3 years. [1] [SEP]

The present manuscript, however, would fit much better in a specialised technical journal than on a

broad arena such as Nature Communications.

Reviewer #3:

Remarks to the Author:

The paper presents a numerical study of the properties of a simple magnetohydrodynamic (fluid) plasma as a topological material, where the topology and a number of relations in a special phase space predicts physical properties of the existence and scope of electron plasma waves. This is a new and rapidly developing area, especially for continuum fluid systems such as conventional fluids and plasmas.

The main results illustrate two plasma cases in detail, using numerical solution, and show that the theoretical topological relationships are satisfied. They primarily extend and complete two cases previously studied by other authors (Refs. 8 and 10), using essentially the same plasma model and approximations.

The paper is clearly written. The results appear correct and are connected to known theory, although the theoretical entities is not described in detail.

It is useful to have complete detailed solutions for basic example cases that can be easily analyzed.

However, the model cases are very limited compared to real plasmas. The 'comprehensive' picture of a magnetized cold plasma (as described in the abstract) actually describes the linearized ideal magnetohydrodynamic equations for a cold plasma in a straight, uniform magnetic field, where the ions do not move. In one case, a 1D background density gradient across the field is used to model a straight plasma cylinder surrounded by vacuum. These simplifications allow the existence of mathematical symmetries that lead to simple topological constraints on the existence and properties of the plasma electron waves. While these and similar systems have been suggested to apply to specific real plasmas, where they are likely to be tested in the near future (see references), and the practical effects of the simplifications and the role of exact symmetries for the topological properties are a subject of active research, the simplifying assumptions are important to interpreting the results. The forced breaking of similar simplifications has seriously handicapped the ability of a number of other types of plasma models to describe real plasmas and limited their practical applicability.

The paper is acceptable for publication with small revisions.

The abstract and conclusions should clearly state the limitations of the plasma model and that the number of topological regions (10) depends on the model. The paper should also clarify that it extends previous solutions and explain what the new results are.

On a minor note, a few typos should be corrected.

It would also help the plasma audience to define the term 'edge' mode as arising from boundaries in topological space rather than on/near the physical boundary of the plasma, which could they also be.

REPLY TO REVIEWER 1

We are very grateful to the reviewer for the positive overall assessment of the manuscript and the constructive comments and suggestions. The following are our replies and specific improvements that have been made in the revised manuscript according to these comments and suggestions.

Reviewer comment 1

“I really feel that this paper will influence thinking in the field, particularly it will renew the sixty years old presentation of the CMA diagram providing a crude topological classification of the modes transition in the density/magnetic field space, but on the restricted phenomenological basis of the visual transition between the wave normal surfaces (roughly speaking transition between spheroid, lemniscoid and hyperboloid), here we have at hand a new topological index of the Chern type providing a predictive tool with respect to the occurrence of edge modes.”

Reply and revision:

We thank the reviewer for raising this important point, and we share the same opinion. In the first section of the revised manuscript, we have included a brief discussion on this connection suggested by the reviewer. The relevant text reads:

“In plasma physics, the well-known Clemmow-Mullaly-Allis (CMA) [16-19] diagram provides a crude topological classification of wave normal surfaces in the parameter space of magnetic field and density. The classification of the waves using the topological index of the Chern type provides a new theoretical understanding that can be used as predictive tool, e.g., in the study of edge modes.”

Reviewer comment 2

“Beside the bulk mode topological classification, the edge modes identified and analyzed here are interesting both from the point of view of theoretical and applied plasma physics and will clarify the Dyakonov type surface waves classification at the interface between magnetized plasmas.”

Reply and revision:

We thank the reviewer for giving this comment. Although we are not experts on surface plasma waves, we added the following short discussion in the Discussion setion of manuscript to point out these studies of surface waves in the past. We hope this can stimulate the interest of the community on this topic.

“The linear and nonlinear properties of edge modes, or surface waves, in plasmas have been studied for decades [28-30]. However, the relation between surface modes and the topology of bulk plasma was only pointed out recently. In the present research, we studied the linear eigenmodes of a cold plasma in a uniform magnetic field, and found that such a magnetized cold plasma has 10 different topological phases in the (ω_p, Ω, k_z) space.”.

Overall, we thank the reviewer again for the careful review of the manuscript and for the constructive suggestions and comments, all of which have been incorporated in the revised manuscript. We trust the reviewer will find the revised manuscript satisfactory.

Yichen Fu

Hong Qin

REPLY TO REVIEWER 2

We thank the reviewer for the careful review of the manuscript and for the the positive overall assessment. We are also grateful to the reviewer for the constructive comments and suggestions, according to which the manuscript has been revised. The following are our replies and revisions.

Reviewer comment 1

“I do not share the authors viewpoint that proper Brillouin zones are needed for using Chern numbers to predicting boundary states. Landau levels are also described in the continuum and the quantum Hall effect is in fact the best established physical application of Chern numbers. Similarly the Chern number of the nodes in Weyl semimetals are related to the Fermi arcs on the boundary and there is no problem describing these phenomena in the continuum.”

Reply and revision:

We thank the reviewer for raising this important issue. In fact, we agree with the reviewer that proper Brillouin zones are not essential for using Chern numbers to predicting boundary states. As the reviewer stated, Landau levels in the continuum and Weyl semimetals are certainly good examples. Furthermore, we recognize the following additional arguments to justify this viewpoint.

- To obtain a compact Brillouin zone, we used a regularization procedure in the k_{\perp} -space for $k_{\perp} \rightarrow \infty$. However, the edge modes we discovered (Fig. 4) are found at relatively small and finite k_{\perp} . It is reasonable to suggest that the property of the system at finite k should not sensitively dependent on its behavior at infinitely large k .
- In a non-compact space, the Chern number of each band might not be an integer. On the other hand, when different bands touch, the change of the Chern number for each band is always an integer (Chapter 2.3 in Ref. [12]). Therefore, the number of edge modes, predicted by the difference of the gap Chern numbers, is still expected to be an integer.

Nevertheless, we feel that it is probably at least helpful to mention the regularization procedure because it is an issue that has been debated about in the research community. Some recent studies have suggested that violations of the bulk-edge correspondence exist in magnetized plasmas, which has been attributed to the lack of a proper Brillouin zone. For example, Gangaraj and Monticone [25] discussed the violation of the bulk-edge correspondence in the gap between two bands, one of which has a non-integer Chern number. They showed that an unphysical edge mode exists at $k \rightarrow \infty$ and argued that the unusual phenomenon will not happen if the Brillouin zone is compact.

In the revised manuscript, we have revised the discussion on this issue in the first section as follows.

“On the other hand, continuum media, including plasmas, do not have well defined Brillouin zones, and compactification techniques in the wavenumber space [4,5,10,14] have been adopted to generate integer Chern numbers. It has also been argued that proper Brillouin zones are not essential for using Chern numbers to predicting boundary states [15]. Landau levels in the continuum and Weyl semimetals are good examples.”

Reviewer comment 2

“The authors conclude: ‘No violation of bulk-edge correspondence is observed in this system.’ This is more of a sanity check as the system is neither driven nor non-Hermitian. ”

Reply and revision:

We agree with the reviewer that in general the bulk-edge correspondence holds in a non-driven Hermitian system. However, as mentioned above, recent studies have reported violations of bulk-edge correspondence in the non-driven Hermitian system of a magnetized plasma [25]. Given this pretext, we feel that it is beneficial to the community to report the confirmation of the bulk-edge correspondence in our study of the non-driven Hermitian system of a magnetized plasma . In response to the reviewer’s comment, we have revised this statement as

“The validity of the bulk-edge correspondence is confirmed for the cold magnetized plasma as a non-driven Hermitian system.”

Reviewer comment 3

“In fact, I would have hoped that authors would have started their exploration where they instead finish the discussion of future directions, namely with the study of non-Hermitian topological phases in plasmas. In that context there is still a lot to learn and explore despite exciting progress the past 3 years.”

Reply and revision:

We agree with the reviewer that further exploration of the non-Hermitian topological phases in plasmas would be exciting and rewarding. However, intrigued by the recent work by Parker et al. [10] and Gangaraj and Monticone [25], we find that there are fundamental and elementary physical issues to be clarified, even in the Hermitian case. This is to a large degree due to the complexity of the magnetized plasma itself. For example, we find that there are ten different topological phases in the parameter space of magnetized plasmas and identified a necessary and sufficient conditions for the existence of surface modes. We believe these contributions are valuable for the community of plasma physics and other related fields. To explain this nature of the study reported, we stated in the Discussion section of the revised manuscript that

“(t)hese results improved our understanding of the elementary properties of the magnetized plasma as a topological material.”

Finally, we would like to express our gratitude to the reviewer again for pointing out places for improvement. We have tried our best to address all the issues raised by the reviewer and revised the manuscript accordingly. These revision has results in important improvement of the manuscript, and we hope that the reviewer will find the revised submission satisfactory

and suitable.

Yichen Fu

Hong Qin

REPLY TO REVIEWER 3

We are very grateful to the reviewer for the positive overall assessment of the manuscript and for identifying opportunities for improvement. In what follows, we address the reviewer's comments and discuss revisions that have been made in the second submission.

Reviewer comment 1

“The paper is clearly written. The results appear correct and are connected to known theory, although the theoretical entities is not described in detail. It is useful to have complete detailed solutions for basic example cases that can be easily analyzed.”

Reply and revision:

We are grateful to the review for the suggestion to provide more detailed information. In response, we have revised the Methods section and clarified the connection between the contents in the Results section and the detailed derivations provided in the Methods section. The relevant text in the Results section now reads:

“..... where $H(\mathbf{k})$ is a 9×9 Hermitian matrix, and $|\psi\rangle = (\mathbf{v}, \mathbf{E}, \mathbf{B})^T$ is a nine-dimensional vector consisting of the perturbed velocity and electromagnetic fields. A detailed derivation of $H(\mathbf{k})$ is provided in the Methods section.”

“..... the system has 9 eigenvalues ω_n and eigenvectors $|\psi_n\rangle$ as functions of \mathbf{k}_\perp , where $\omega_{-n} = -\omega_n$ and $n = -4, -3, \dots, 3, 4$ is the index for the eigenmodes. See the Methods section for a detailed discussion of the symmetry properties of the spectrum.”

“..... Chern numbers admit the following symmetries, $C_n(\Omega, k_z) = C_n(\Omega, -k_z) = -C_n(-\Omega, k_z) = -C_{-n}(\Omega, k_z)$. See additional details about the symmetry properties of the Chern numbers for the system in the Methods section.”

“To ensure the numerically calculated spectrum admits the same symmetries as the analytical spectrum, we adopt a discretization scheme that preserves the Hermiticity and the particle-hole symmetry of the system. See the Methods section for details.”

The revised discussion in the Methods section on the numerical algorithm adopted now reads:

“Equations (14)-(16) are now a standard matrix eigenvalue problem, which can be solved by various established algorithms. It is straightforward to confirm that the matrix specified by Eqs. (14)-(16) is Hermitian and admits the particle-hole symmetry as Eq. (10) is and does. This structure-preserving discretization enables the reformulation of the eigenvalue problem of an inhomogeneous, magnetized, cold plasma as an eigenvalue problem for a Hermitian matrices with the particle-hole symmetry. ”

Reviewer comment 2

“However, the model cases are very limited compared to real plasmas. The ‘comprehensive’ picture of a magnetized cold plasma (as described in the abstract) actually describes the linearized ideal magnetohydrodynamic equations for a cold plasma in a straight, uniform magnetic field, where the ions do not move. In one case, a 1D background density gradient across the field is used to model a straight plasma cylinder surrounded by vacuum. These simplifications allow the existence of mathematical symmetries that lead to simple topological constraints on the existence and properties of the plasma electron waves. While these and similar systems have been suggested to apply to specific real plasmas, where they are likely to be tested in the near future (see references), and the practical effects of the simplifications and the role of exact symmetries for the topological properties are a subject of active research, the simplifying assumptions are important to interpreting the results. The forced breaking of similar simplifications has seriously handicapped the ability of a number of other types of plasma models to describe real plasmas and limited their practical applicability.”

Reply and revision:

We thank the reviewer for pointing out the limit and value of the classification study carried out in the present study, and we agree with this assessment. In the revised manuscript, we have revised the Discussion section to clearly state these viewpoints. The relevant text now reads:

“The present classification of the topological phases in the parameter space is carried out for the linearized system of a homogeneous, magnetized, cold plasma with stationary ions. Such a classification in this simplified system provides an elementary tool and serves as a reference model for studying the topological-matter properties of more realistic plasmas. For example, in an inhomogeneous plasma, edge modes may be excited at the interface between two plasmas in different topological phases. When other physical effects, such finite temperature, plasma collisions, and kinetic interactions, are important, the band structures [16] and topological properties of the system can change significantly, including the number of topological phases and possible phase transitions. One systematic approach for understanding these changes is to analyze the variations of the symmetry properties of the system [34]. For instance, the magnetized cold plasma model studied here has a broken time-reversal symmetry, but the linearized ideal magnetohydrodynamics system is invariant under the (modified) time-reversal transformation [8], despite the existence of an external magnetic field. Furthermore, the linear dynamics in many plasma models is expected to be non-Hermitian [35, 36], permitting unstable and damped eigenmodes. Applying the methods of topological phases for non-Hermitian systems [37, 38] will bring new insights and discoveries in the study of plasma instabilities for laboratory and astrophysical plasmas. ”

Reviewer comment 3

“The abstract and conclusions should clearly state the limitations of the plasma model and that the number of topological regions (10) depends on the model.”

Reply and revision:

We thank the reviewer for this suggestion. In the Abstract, the first section, and the Discussion section of the revised manuscript, we have explicitly stated what simplified plasma model is used and discussed the dependence of the number of topological phases on the plasma model adopted. In the Abstract, the relevant text now reads:

“We find that for the linear eigenmodes, there are 10 topological phases...”

In the first section, the relevant text now reads:

“We extend the study by Parker et al. [10] of the linear eigenmodes in a cold plasma with stationary ions in a uniform magnetic field, and carefully draw the topological phase diagram in the parameter space, and clarify the condition for the existence of the topological edge modes.”

In the Discussion section, the relevant text now reads:

“In the present research, we studied the linear eigenmodes of a cold plasma in a uniform magnetic field, and found that the system has 10 different topological phases in the (ω_p, Ω, k_z) space.”

“The present classification of the topological phases in the parameter space is carried out for the linearized system of a homogeneous, magnetized, cold plasma with stationary ions. Such a classification in this simplified system provides an elementary tool and serves as a reference model for studying the topological-matter properties of more realistic plasmas. For example, in an inhomogeneous plasma, edge modes may be excited at the interface between two plasmas in different topological phases. When other physical effects, such finite temperature, plasma collisions, and kinetic interactions, are important, the band structures [16] and topological properties of the system can change significantly, including the number of topological phases and possible phase transitions.”

As stated by the reviewer, the symmetry property of bulk modes also depends on the choice of plasma models. Therefore we have added the following statements in the Discussion section as well:

“One systematic approach for understanding these changes is to analyze the variations of the symmetry properties of the system [34]. For instance, the magnetized cold plasma model studied here has a broken time-reversal symmetry, but the linearized ideal magnetohydrodynamics system is invariant under the (modified) time-reversal transformation [8], despite the existence of an external magnetic field. Furthermore, the linear dynamics in many plasma models is

expected be to non-Hermitian [35, 36], permitting unstable and damped eigenmodes. Applying the methods of topological phases for non-Hermitian systems [37, 38] will bring new insights and discoveries in the study of plasma instabilities for laboratory and astrophysical plasmas.”

Reviewer comment 4

“The paper should also clarify that it extends previous solutions and explain what the new results are.”

Reply and revision:

We thank the reviewer for this suggestion. In the revised manuscript, we have clarified that we extend the work by Parker et al. [10] and highlighted the new contribution of our investigation. In particular, we emphasized the following new findings in the manuscript. 1) The magnetized cold plasma model has 10 topological phases. 2) A necessary and sufficient condition for the existence of edge mode is derived. 3) The edge mode exists not only in the plasma-vacuum boundary as reported in Ref. [10], but also in more general plasma-plasma boundaries. We have revised the text in the first section and the relevant text now reads:

“We extend the study by Parker et al. [10] of the linear eigenmodes in a cold plasma with stationary ions in a uniform magnetic field, and carefully draw the topological phase diagram in the parameter space, and clarify the condition for the existence of the topological edge modes. Specifically, we report the following findings. i) In the parameter space of magnetic field $\mathbf{B} = B_0\hat{z}$, density n , and wavenumber k_z , there are 10 topological phases, separated by the Langmuir wave-L wave (LL) resonance, the Langmuir wave-Cyclotron wave (LC) resonance, and the $B = 0$ surface. ii) Their topological properties are classified by the integer Chern numbers of the spectrum. For fixed non-vanishing B and k_z , there are two possible topological phase transitions due to the two resonances, while only the transition at the LC resonance produces edge modes. iii) There

exists a critical density n_c such that plasmas below and above n_c are in different topological phases across the LC resonance. We find that edge modes exist not only at the plasma-vacuum boundary [10], but also at more general plasma-plasma interfaces when a necessary and sufficient condition, Eq. (7), is satisfied, and the edge modes can be categorized by different behaviors of the Fermi arcs or Fermi-arcs-like curves. This finding broadens the possible applications of these exotic edge modes in space and laboratory plasmas.”

Reviewer comment 5

“On a minor note, a few typos should be corrected.”

Reply and revision:

We thank the reviewer for carefully reading the manuscript. In the revised manuscript, handful typos have been corrected.

Reviewer comment 6

“It would also help the plasma audience to define the term ‘edge’ mode as arising from boundaries in topological space rather than on/near the physical boundary of the plasma, which could they also be.”

Reply and revision:

We appreciate that the reviewer raised this point. Indeed the meaning of ‘edge’ needs to be clarified since ‘edge’ usually means the physical boundary of plasma, especially in the fusion plasma community. We have added the following statements in the Discussion section:

“It is worth mentioning that in the plasma physics community, the term ‘edge’ usually refers to the physical boundary of plasmas adjacent to the first wall of vacuum chambers [31,32]. In the context of topological matters, the term ‘edge’ refers to the boundary of two topologically different regions, although a plasma-vacuum edge can also be a topological edge when the plasma and the vacuum have different Chern numbers. As discussed above, the topological edges include not only the plasma-vacuum boundary, but also more general gaseous plasma-plasma interfaces, which are the focus of the present study.”

Overall, we thank the reviewer again for the valuable comments and suggestions, all of which have been adopted in the revised manuscript. These revisions in response to the reviewer’s comments and suggestions have resulted in significant improvement of the manuscript, and we trust that the reviewer will find the revised manuscript satisfactory.

Yichen Fu
Hong Qin

Reviewers' Comments:

Reviewer #3:

Remarks to the Author:

The authors have done a good job in carefully addressing the comments made on the first version of the paper. The revised version is much improved. It explains more clearly the relevance of the topological phase analysis to a simple plasma, in a way that presents a better conceptual basis for developing extensions to more complex and more realistic plasmas.

As such, the paper is suitable for publication in Nature Communications.